# A Theory of Optimistically Universal Online Learnability for General Concept Classes

**Steve Hanneke**
Department of Computer Science
Purdue University
West Lafayette, IN 47907
steve.hanneke@gmail.com

**Hongao Wang**
Department of Computer Science
Purdue University
West Lafayette, IN 47907
wang5270@purdue.edu

## Abstract

We provide a full characterization of the concept classes that are optimistically universally online learnable with $\{0, 1\}$ labels. The notion of optimistically universal online learning was defined in [Hanneke, 2021] in order to understand learnability under minimal assumptions. In this paper, following the philosophy behind that work, we investigate two questions, namely, for every concept class: (1) What are the minimal assumptions on the data process admitting online learnability? (2) Is there a learning algorithm which succeeds under every data process satisfying the minimal assumptions? Such an algorithm is said to be optimistically universal for the given concept class. We resolve both of these questions for all concept classes, and moreover, as part of our solution we design general learning algorithms for each case. Finally, we extend these algorithms and results to the agnostic case, showing an equivalence between the minimal assumptions on the data process for learnability in the agnostic and realizable cases, for every concept class, as well as the equivalence of optimistically universal learnability.

## 1 Introduction

Just as computability is a core question in computation theory, learnability is now a core question in learning theory. Intuitively, learnability is trying to ask whether we can predict the future correctly with high probability by observing enough examples. In order to describe this intuition formally, we need to define learning models, such as, Probably Approximately Correct (PAC) learning (Vapnik and Chervonenkis [1974] and Valiant [1984]), online learning (Littlestone and Warmuth [1986]) and query learning (Angluin [1988]). In this paper, we focus on a variant of online learning: online learning under data processes. In this setting, the learning is sequential: in round $t$, an instance $X_t$ is given to the algorithm, and then the algorithm makes the prediction, $\hat{Y}_t$, based on the history $(X_{\leq t-1}, Y_{\leq t-1})$ and the input $X_t$, i.e., $\hat{Y}_t = f_t(X_{\leq t-1}, Y_{\leq t-1}, X_t)$. Next, the target value $Y_t$ will be revealed to the learner such that it can be used to inform future predictions. We model this sequence as a general *stochastic process* $(\mathbb{X}, \mathbb{Y}) = \{(X_t, Y_t)\}_{t \in \mathbb{N}}$ (possibly with dependencies across times $t$). We say that the algorithm is strongly consistent under $(\mathbb{X}, \mathbb{Y})$ if the long-run average error is guaranteed to be low, i.e., $\frac{1}{n} \sum_{t=1}^{n} \mathbb{I}\left[\hat{Y}_t \neq Y_t\right] \to 0$ almost surely, when $n \to \infty$.

In our setting, any theory of learning must be expressed based on the properties of, or restriction on, the data process, as the mistake bound is based on the data process. Thus, following an approach found in much of the learning theory literature, such as the PAC model of Valiant [1984] and Vapnik and Chervonenkis [1971] and the online learning model of Littlestone [1988], we introduce the restriction by an additional component, concept class $\mathcal{H} \subseteq \mathcal{Y}^{\mathcal{X}}$. The role of the concept class is to restrict the processes we need to face, such that they all are realizable under that concept class.

If there is a target function $h^* \in \mathcal{H}$, such that $Y_t = h^*(X_t)$ for every $t$, we say the data process $(\mathbb{X}, \mathbb{Y})$ is realizable (though our formal definition below is slightly more general). For a given $\mathbb{X}$, if a learning rule is strongly consistent under $(\mathbb{X}, \mathbb{Y})$ for *every* $\mathbb{Y}$ such that $(\mathbb{X}, \mathbb{Y})$ is realizable, we say it is *universally consistent* under $\mathbb{X}$ in the realizable case.

It is known that we cannot get the low average error guarantee for all concept classes and data processes [Hanneke, 2021]. Thus, we should make several restrictions on either the data process, the concept class, or a mixture of both. All three types of restrictions have been investigated: Littlestone [1988], Ben-David et al. [2009] studied online learning with unrestricted data processes but restricted concept classes. Haussler et al. [1994], Ryabko [2006] researched the online learning problem with a mix of both restrictions. There are also substantial amount of papers investigating online learnability with all measurable functions but restricted data processes. Most of these specifically consider the case of i.i.d. processes, such as Stone [1977], Devroye et al. [1996], though this has also been generalized to general stationary ergodic processes [Morvai et al., 1996, Gyorfi et al., 1999] or processes with certain convergence properties enabling laws of large numbers [Morvai et al., 1999, Steinwart et al., 2009].

More recently, a general question has been studied: In the case of $\mathcal{H}$ equals the set of all measurable functions, is there a learning rule guaranteed to be universally consistent given only the assumption on $\mathbb{X}$ that universally consistent learning is possible under $\mathbb{X}$? The assumption that universal consistency is possible under $\mathbb{X}$ is referred to as the "optimist's assumption" [Hanneke, 2021], and for this reason, learning rules which are universally consistent for all $\mathbb{X}$ satisfying the optimist's assumption are said to be *optimistically universally consistent*. There is a series of works focusing on this question, starting from Hanneke [2021] and continuing with Blanchard and Cosson [2022], Blanchard [2022], Blanchard et al. [2022a]. They tackle this question by first characterizing the minimal assumptions on the data process admitting universally consistent learning and then proposing an online learning algorithm that is universally consistent for all data processes satisfying that assumption. However, their works all focus on the situation with no restrictions on concepts in the concept class (i.e., $\mathcal{H}$ as all measurable functions). Thus, a natural question arises: For which concept classes do there exist optimistically universal learning rules?

In this paper, we investigate the question mentioned above when the output is binary, i.e. $\{0, 1\}$. We handle this problem by first figuring out the minimal assumptions on the data process admitting consistent online learning as well. Thus, our results answered that question in the following aspects:

- For which concept classes is optimistically universally consistent learning possible?
- What are the sufficient and necessary conditions for processes to admit universally consistent online learning for a given concept class $\mathcal{H}$?
- For which concept classes is it the case that *all* processes $\mathbb{X}$ admit universally consistent online learning?

We first answer these questions in the realizable case. Surprisingly, the answers turn out to be intimately related to combinatorial structures arising in the work of Bousquet et al. [2021] on universal learning rates. This suggests a potential connection between the universal consistency of online learning and universal learning rates, which is of independent interest. We also extend our learning algorithms for the realizable case to the agnostic setting, where the requirement of low average loss is replaced by that of having sublinear regret. Interestingly, our answers to the above questions remain unchanged in the agnostic case, establishing an equivalence between agnostic and realizable learning in this setting.

In this paper, we first provide some interesting examples in section 3. Then section 4 investigates question three and question one for those classes and section 5 answers question two and question one for remaining classes. Finally, in section 6, we extend our algorithm to the agnostic case.

## 1.1 More Related Work

Starting from Littlestone's ground-breaking work [Littlestone, 1988], online learning is becoming more and more important. In this paper, Littlestone [1988] introduces a combinatorial parameter of the concept class, known as Littlestone dimension, to characterize the online learnable concept classes in the realizable case. After that, Ben-David et al. [2009] figure out Littlestone dimension is still the property to characterize online learnability in the agnostic setting. They extend an online

learning algorithm for the realizable case to such an algorithm for the agnostic case using the weighted majority algorithm from Littlestone and Warmuth [1994]. This line of work makes no assumption on the data process and investigates how restrictions on the concept affect the learnability of the problem. There are two other categories of assumptions also investigated in history: one is to make assumptions on both the data process and the concept, and the other is to make assumptions on the data process but not the concept. Those two categories are discussed in detail subsequently.

First, the works investigating the question of learnability with restrictions on both the data process and the concept make a variety of assumptions. For example, Haussler et al. [1994] investigate how the restrictions on the concept will affect learnability given that the data process is i.i.d. This problem is more similar to a streamlined version of PAC learning and they show that there is a logarithmic mistake bound with the assumption that the data process is i.i.d. and the concept class has finite VC dimension. Adams and Nobel [2010] reveal that all stationary ergodic sequences will uniformly converge under the concept class with finite VC dimension. However, they cannot show the convergence rate of that learning algorithm. Many other works focus on revealing the rate with slightly stronger assumptions on the sequences, such as, [Yu, 1994, Karandikar and Vidyasagar, 2002].

Another line of works focuses on the question of learnability with restrictions on the process instead of the concept, starting from the theory of universally consistent predictors under i.i.d sequences. In particular, there exists an online learning rule, such that for any i.i.d. sequence $(\mathbb{X}, \mathbb{Y})$, and every measurable function $f^*$, the long-run average loss is 0 almost surely, such as, [Stone, 1977, Devroye et al., 1996, Hanneke et al., 2021]. In the meanwhile, people are also interested in the consistency under non-i.i.d. processes; Gyorfi et al. [1999], Morvai et al. [1996] reveal that there are universally consistent online learning rules under general stationary ergodic processes. The paper of Morvai et al. [1999] and the paper of Steinwart et al. [2009] show universal consistency under some classes of processes satisfying laws of large numbers.

More recently, the work of Hanneke [2021] investigates whether there is a consistent learning rule given only the assumptions on $\mathbb{X}$ that universally consistent learning is possible. This work generalizes the assumptions on $\mathbb{X}$ made by the previous works on universal consistency. The assumption that universally consistent learning is possible is known as the "optimist's assumption", so the consistency under that assumption is called *optimistically universal* consistency. Hanneke [2021] studies three different learning models: inductive, self-adaptive, and online, and proves that there is an optimistically universal self-adaptive learning rule and no optimistically universal inductive learning rule. After this beginning, the works of Blanchard and Cosson [2022], Blanchard et al. [2022a], Blanchard [2022] show that optimistically universal online learning is possible and the processes that admit strongly universal online learning satisfy the condition called $\mathcal{C}_2$ (see condition C for reference). This problem is also investigated under different models, such as, in contextual bandit setting Blanchard et al. [2022b, 2023] and general noisy labels Blanchard and Jaillet [2023].

## 2 Preliminaries and Main Results

In this section, we provide formal definitions and model settings and briefly list the main results of this paper without proof. For brevity, we provide the high-level proof sketch in the subsequent sections and proof details are in the appendices.

**Model Setting** We formally provide the learning model here. Let $(\mathcal{X}, \mathcal{B})$ be a measurable space, in which $\mathcal{X}$ is assumed to be non-empty and $\mathcal{B}$ is a Borel $\sigma$-algebra generated by a separable metrizable topology $\mathcal{T}$. We also define a space $\mathcal{Y} = \{0, 1\}$ called *label space*. Here we focus on learning under the 0-1 loss: that is, $(y, y') \mapsto \mathbb{I}[y \neq y']$ defined on $\mathcal{Y} \times \mathcal{Y}$, where $\mathbb{I}[\cdot]$ is the indicator function. A stochastic process $\mathbb{X} = \{X_t\}_{t \in \mathbb{N}}$ is a sequence of $\mathcal{X}$-valued random variables. A stochastic process $\mathbb{Y} = \{Y_t\}_{t \in \mathbb{N}}$ is a sequence of $\{0, 1\}$-valued random variable. The concept class $\mathcal{H}$, which is a non-empty set of measurable functions $h : \mathcal{X} \to \mathcal{Y}$.[1]

The online learning rule is a sequence of measurable functions: $f_t : \mathcal{X}^{t-1} \times \mathcal{Y}^{t-1} \times \mathcal{X} \to \mathcal{Y}$, where $t$ is a non-negative integer. For convenience, we also define $\hat{h}_{t-1} = f_t(X_{<t}, Y_{<t})$, here $(X_{<t}, Y_{<t}) = \{(X_i, Y_i)\}_{i<t}$ is the history before round $t$.

---

[1] We additionally make standard restrictions on $\mathcal{H}$ to ensure certain estimators are well-behaved; we omit the details for brevity, but refer the reader to Bousquet et al. [2021] for a thorough treatment of measurability of these estimators.

There are two ways to define the realizable case: The most common one is that there exists $h^* \in \mathcal{H}$ such that $Y_t = h^*(X_t)$. The other is the definition 1 on the realizable data process, which comes from the universal learning setting. These two definitions are equivalent in the uniform PAC learning with i.i.d. samples. However, they are different when talking about universal learning. Thus, we follow the definition from the universal learning setting.

**Definition 1.** *For every concept class $\mathcal{H}$, we can define the following set of processes $R(\mathcal{H})$:*

$$R(\mathcal{H}) := \left\{ (\mathbb{X}, \mathbb{Y}) = \{(X_i, Y_i)\}_{i \in \mathbb{N}} : \text{with probability } 1, \forall n < \infty, \{(X_i, Y_i)\}_{i \leq n} \text{ realizable by } \mathcal{H} \right\}.$$

In the same way, the set of realizable label processes:

**Definition 2.** *For every concept class $\mathcal{H}$ and data process $\mathbb{X}$, define a set $R(\mathcal{H}, \mathbb{X})$ of label processes:*

$$R(\mathcal{H}, \mathbb{X}) := \left\{ \mathbb{Y} = \{Y_i\}_{i \in \mathbb{N}} : (\mathbb{X}, \mathbb{Y}) \in R(\mathcal{H}) \text{ and } \exists \text{ a non-random function } f \text{ s.t. } Y_i = f(X_i) \right\}.$$

In other words, $R(\mathcal{H}, \mathbb{X})$ are label processes $\mathbb{Y} = f(\mathbb{X})$ s.t. $(\mathbb{X}, f(\mathbb{X})) \in R(\mathcal{H})$. Importantly, while every $f \in \mathcal{H}$ satisfies $f(\mathbb{X}) \in R(\mathcal{H}, \mathbb{X})$, there can exist $f \notin \mathcal{H}$ for which this is also true, due to $R(\mathcal{H})$ only requiring realizable *prefixes* (thus, in a sense, $R(\mathcal{H}, \mathbb{X})$ represents label sequences by functions in a *closure* of $\mathcal{H}$ defined by $\mathbb{X}$).[2]

At first, we define the universal consistency under $\mathbb{X}$ and $\mathcal{H}$ in the realizable case. An online learning rule is universally consistent under $\mathbb{X}$ and $\mathcal{H}$ if its long-run average loss approaches $0$ almost surely when the number of rounds $n$ goes to infinity for all realizable label processes. Formally, we have the following definition:

**Definition 3.** *An online learning rule is* strongly universally consistent *under $\mathbb{X}$ and $\mathcal{H}$ for the realizable case, if for* every $\mathbb{Y} \in R(\mathcal{H}, \mathbb{X})$, $\limsup_{n \to \infty} \frac{1}{n} \sum_{t=1}^{n} \mathbb{I}\left[Y_t \neq \hat{h}_{t-1}(X_t)\right] = 0$ a.s.

We also define the universal consistency under $\mathbb{X}$ and $\mathcal{H}$ for the agnostic case. In that definition, we release the restrictions that $\mathbb{Y} \in R(\mathcal{H}, \mathbb{X})$, instead the label process $\mathbb{Y}$ can be set in any possible way, even dependent on the history of the algorithm's predictions. Thus, the average loss may be linear and inappropriate for defining consistency. Therefore, we compare the performance of our algorithm with the performance of the best possible $\mathbb{Y}^* \in R(\mathcal{H}, \mathbb{X})$, which is usually referred to as *regret*. We say an online algorithm is universally consistent under $\mathbb{X}$ and $\mathcal{H}$ for the agnostic case if its long-run average regret is low for every label process. Formally,

**Definition 4.** *An online learning rule is* strongly universally consistent *under $\mathbb{X}$ and $\mathcal{H}$ for the agnostic case, if for* every $\mathbb{Y}^* \in R(\mathcal{H}, \mathbb{X})$ *and for* every $\mathbb{Y}$, $\limsup_{n \to \infty} \frac{1}{n} \sum_{t=1}^{n} \left( \mathbb{I}\left[Y_t \neq \hat{h}_{t-1}(X_t)\right] - \mathbb{I}[Y_t \neq Y_t^*] \right) \leq 0$ a.s.

To describe the assumption that universal consistency is possible under the data process $\mathbb{X}$ and the concept class $\mathcal{H}$, we need to define the process admitting universal online learning as follows:

**Definition 5.** *We say a process $\mathbb{X}$ admits* strongly universal online learning *(or just* universal online learning *for convenience) if there exists an online learning rule that is strongly universally consistent under $\mathbb{X}$ and $\mathcal{H}$.*

If the online learning rule is universally consistent under every process that admits universal online learning, we call it **optimistically universal** under the concept class. If there is an optimistically universal learning algorithm under that concept class, we say that concept class is optimistically universally online learnable. The formal definition is provided below:

**Definition 6.** *An online learning rule is* optimistically universal *under concept class $\mathcal{H}$ if it is strongly universally consistent under every process $\mathbb{X}$ that admits strongly universally consistent online learning under concept class $\mathcal{H}$.*

*If there is an online learning rule that is* optimistically universal *under concept class $\mathcal{H}$, we say $\mathcal{H}$ is* optimistically universally online learnable.

---

[2]For instance, for $\mathcal{X} = \mathbb{N}$, for the process $X_i = i$, and for $\mathcal{H} = \{\mathbb{1}_{\{i\}} : i \in \mathcal{X}\}$ (singletons), the all-0 sequence is in $R(\mathcal{H}, \mathbb{X})$ though the all-0 function is not in $\mathcal{H}$.

Next, we define the combinatorial structures we use to characterize the concept class that makes all processes admit universal online learning and is optimistically universally online learnable when all processes admit strongly universally consistent online learning:

**Definition 7** (Littlestone tree Bousquet et al. [2021]). *A Littlestone Tree for $\mathcal{H}$ is a complete binary tree of depth $d \leq \infty$ whose internal nodes are labeled by $\mathcal{X}$, and whose two edges connecting a node to its children are labeled $0$ and $1$, such that every finite path emanating from the root is consistent with a concept $h \in \mathcal{H}$. We say that $\mathcal{H}$ has **an infinite Littlestone tree** if it has a Littlestone tree of depth $d = \infty$.*

**Definition 8** (VCL Tree Bousquet et al. [2021]). *A **VCL Tree** for $\mathcal{H}$ of depth $d \leq \infty$ is a collection*

$$\{x_u \in \mathcal{X}^{k+1} : 0 \leq k < d, u \in \{0,1\}^1 \times \{0,1\}^2 \times \cdots \times \{0,1\}^k\}$$

*such that for every $n < d$ and $y \in \{0,1\}^1 \times \{0,1\}^2 \times \cdots \times \{0,1\}^{n+1}$, there exists a concept $h \in \mathcal{H}$ so that $h(x^i_{y_{\leq k}}) = y^i_{k+1}$ for all $0 \leq i \leq k$ and $0 \leq k \leq n$, where we denote*

$$y_{\leq k} = (y^0_1, (y^0_2, y^1_2), \ldots, (y^0_k, \ldots, y^{k-1}_k)), x_{y_{\leq k}} = (x^0_{y_{\leq k}}, \ldots, x^k_{y_{\leq k}})$$

*We say that $\mathcal{H}$ has **an infinite VCL tree** if it has a VCL tree of depth $d = \infty$.*

The characterization is formally stated in the following two theorems:

**Theorem 9.** *If and only if a concept class $\mathcal{H}$ has no infinite VCL tree, any process admits strongly universally consistent online learning under $\mathcal{H}$.*

**Theorem 10.** *If and only if a concept class $\mathcal{H}$ has no infinite Littlestone tree, any process admits strongly universally consistent online learning under $\mathcal{H}$, and the concept class $\mathcal{H}$ is optimistically universally online learnable.*

According to theorem 9, we know that for those concept classes with an infinite VCL tree, there exist some processes that do not admit universal online learning. Thus, we need to figure out the sufficient and necessary conditions that the processes required to admit universal online learning.

First, we define the experts as algorithms that generate predictions only based on the input $X_t$. Then we define the following condition (which is a property of a data process) and state the main theorem formally:

**Condition A.** *For a given concept class $\mathcal{H}$, there exists a countable set of experts $E = \{e_1, e_2, \ldots\}$, such that $\forall \mathbb{Y}^* \in R(\mathcal{H}, \mathbb{X})$, $\exists i_n \to \infty$, with $\log i_n = o(n)$, such that:*

$$\mathbb{E}\left[\limsup_{n \to \infty} \min_{e_i : i \leq i_n} \frac{1}{n} \sum_{t=1}^{n} \mathbb{I}\left[e_i(X_t) \neq Y_t^*\right]\right] = 0 \tag{1}$$

**Theorem 11.** *A process $\mathbb{X}$ admits strongly universally consistent online learning under concept class $\mathcal{H}$ with infinite VCL tree if and only if it satisfies condition A.*

Next, the sufficient and necessary conditions (on the concept class) for optimistically universal online learning:

**Condition B.** *There exists a countable set of experts $E = \{e_1, e_2, \ldots\}$, such that for any $\mathbb{X}$ admits universal online learning, and any $\mathbb{Y}^* \in R(\mathcal{H}, \mathbb{X})$, there exists $i_n \to \infty$, with $\log i_n = o(n)$, such that:*

$$\mathbb{E}\left[\limsup_{n \to \infty} \min_{e_i : i \leq i_n} \frac{1}{n} \sum_{t=1}^{n} \mathbb{I}\left[e_i(X_t) \neq Y_t^*\right]\right] = 0 \tag{2}$$

Notice that these two conditions (condition A and B) only have one major difference: whether the countable set of experts depends on the process $\mathbb{X}$.

**Theorem 12.** *A concept class $\mathcal{H}$ with infinite VCL tree is optimistically universally online learnable if and only if it satisfies condition B.*

We also extend the algorithms for realizable cases to an algorithm for agnostic cases and show that the same characterization works for agnostic cases.

## 3   Examples

In this section, we provide some interesting examples to help the reader get a sense of what these conditions are. We first provide an example of the concept class that is universally online learnable under all processes but not optimistically universally online learnable.

**Example 1.** *We have the instance space $\mathcal{X} = \mathbb{R}$ and $\mathcal{Y} = \{0, 1\}$, a binary output. The concept class $\mathcal{H}$ is all of the threshold functions. In other words, $\mathcal{H}_{threshold} = \{h_a : h_a(x) = \mathbb{I}[x \geq a] \,|a \in \mathbb{R}\}$. This concept class has no infinite VCL tree, as there is no $(x_1, x_2)$ such that $\mathcal{H}_{threshold}$ shatters all possible results. Thus, all processes admit strongly universally consistent online learning under $\mathcal{H}_{threshold}$. However, it has an infinite Littlestone tree. Thus, for any learning algorithm, there exists a process that is not learnable by that algorithm. So it is not optimistically universally online learnable.*

Referring to that line of optimistically universal online learning papers, we know that the concept class of all measurable functions is optimistically universally online learnable. The sufficient and necessary condition for processes to admit universal online learning under all measurable functions is the condition $\mathcal{C}_2$ (see below). In the meanwhile, our conditions: A and B vanish to $\mathcal{C}_2$ when the concept class $\mathcal{H}$ becomes the class of all measurable functions.

**Condition C** ($\mathcal{C}_2$ in  Hanneke [2021]). *For every sequence $\{A_k\}_{k=1}^{\infty}$ of disjoint elements of $\mathcal{B}$,*

$$|\{k \in \mathbb{N} : X_{1:T} \cap A_k \neq \emptyset\}| = o(T) \ a.s.$$

The following example shows that whether a concept class is optimistically universally online learnable is neither sufficient nor necessary to determine whether its subset is optimistically universally online learnable or not. Whether a concept class is optimistically universally online learnable will be sufficient and necessary to determine whether its subset is optimistically universally online learnable, if and only if the processes that admit universal online learning are the same under those two concept classes.

**Example 2.** *We have the data which is sampled from input space $\mathcal{X} = \mathcal{X}_1 \cup \mathcal{X}_2$ and here $\mathcal{X}_1$ and $\mathcal{X}_2$ are disjoint. For example, $\mathcal{X}_1 = \mathbb{R}^+$ and $\mathcal{X}_2 = \mathbb{R}\backslash\mathbb{R}^+$. Then we can define the concept class: $\mathcal{H}_1$ is the set of all threshold functions on $\mathcal{X}_1$ which are 0 on $\mathcal{X}_2$, and $\mathcal{H}_2$ is a set of all functions on $\mathcal{X}_2$ which are constant on $\mathcal{X}_1$. Then we can consider the following scenarios:*

1. *$\mathcal{H} = \mathcal{H}_2$: It is optimistically universally online learnable. The processes that admit universal online learning will satisfy $\mathcal{C}_2$ if we replace all the $X_t \in \mathcal{X}_1$ as dummy points.*

2. *$\mathcal{H} = \mathcal{H}_1 \cup \mathcal{H}_2$: It is not optimistically universally online learnable, as all processes supported on $\mathcal{X}_1$ admit universal online learning under $\mathcal{H}$. However, for every learning strategy, there exists at least one process on $\mathcal{X}_1$ forcing that strategy to make linear mistakes. (Due to theorem 9 and theorem 10)*

3. *$\mathcal{H}$ are all measurable functions on $\mathcal{X}$. This is also optimistically universally online learnable.*

## 4   Sufficient and Necessary Condition that ALL Processes Admit Universal Online Learning

In this section, we answer the question: *What restrictions on concept classes make ALL processes admit universal online learning under $\mathcal{H}$?* The answer is formally stated as Theorem 9. We show the sufficiency by providing a universal online learning rule (depending on $\mathbb{X}$) under every process $\mathbb{X}$ and $\mathcal{H}$ with no infinite VCL tree.

First, we define the VCL game along with the VCL tree. In this game, there are two players: the learner, $P_L$, and the adversary, $P_A$ and $U_0 = \emptyset$. Then in each round $k$:

- $P_A$ choose the point $X_{(k)} = (X_{k,1}, \ldots, X_{k,k}) \in \mathcal{X}^k$.
- $P_L$ choose the prediction $g_{U_{k-1}}((X_{k,1}, \ldots, X_{k,k})) \in \{0, 1\}^k$.
- Update $U_k = U_{k-1} \cup \{X_{(k)}, g_{U_{k-1}}\}$.
- $P_L$ wins the game in round $k$ if $\mathcal{H}_{U_k} = \emptyset$.

Here $\mathcal{H}_{U_k} = \{h \in \mathcal{H} : \forall i, h(X_{(i)}) = g_i(X_{(i)})\}$, which is the subset of $\mathcal{H}$ that is consistent on $(X_{(i)}, g_i(X_{(i)}))$ for all $i \leq k$.

A *strategy* is a way of playing that can be fully determined by the foregoing plays. And a *winning strategy* is a strategy that necessarily causes the player to win no matter what action one's opponent takes. We have the following lemma from Bousquet et al. [2021].

**Lemma 13** (Bousquet et al. [2021] **lemma 5.2**). *If $\mathcal{H}$ has no infinite VCL tree, then there is a universally measurable winning strategy $g$ for $P_L$ in the game.*

Notice that the winning strategy $g$ is completely decided by $U$, we use $g_U$ as the winning strategy induced by the set $U$. We may use this winning strategy $g_U$ to design an online learning algorithm 1. This algorithm is a combination of the algorithm in the work of Bousquet et al. [2021] and the algorithm inspired by the learning algorithm for partial concept in the work of Alon et al. [2021].

In order to describe the algorithm, we first provide the definitions of partial concepts. A partial concept class $\mathcal{H} \subseteq \{0, 1, *\}^{\mathcal{X}}$ is a set of partial function defined on $\mathcal{X}$, where $h(x) = *$ if and only if $h$ is undefined on $x$. And for a set $X' \subseteq \mathcal{X}$, $X'$ is shattered if every binary pattern $y \in \{0,1\}^{X'}$ is realized by some $h \in \mathcal{H}$. In this algorithm, we have $w(\mathcal{H}', X_{\leq T}) = |\{S : S \subseteq \{x_i\}_{i \leq T}$ such that $S$ shattered by $\mathcal{H}'\}|$, which is the number of the subsequences of the sequence $X_{\leq T}$ that can be shattered by the partial concept class $\mathcal{H}'$. $\mathcal{H}^{g_U} = \{h : \forall X_1, X_2, \ldots, X_k \in \mathcal{X}, (h(X_1), h(X_2), \ldots, h(X_k)) \neq g_U(X_1, X_2, \ldots, X_k)\}$ is the partial concept class induced by $g_U$, which contains the concepts that are not consistent with $g_U$ at more than $k-1$ data points, if $U = \{(X_{(i)}, g_i(X_{(i)}))\}_{i \leq k-1}$. We define $\mathcal{H}^{g_U}_{\{(X_i, Y_i)\}_{i \leq t}} = \{h \in \mathcal{H}^{g_U} : \forall i \leq t, h(X_i) = Y_i\}$. We also define $X_{[t, t']} = \{X_i\}_{t \leq i \leq t'}$ and $t(m) = \frac{m(m+1)}{2}$.

---

**Algorithm 1:** Learning algorithm from winning strategy

$k = 1, U = \{\}, t' \leftarrow 0$.
**for** $t = 1, 2, 3, \ldots$ **do**
    **if** $\exists j_1, j_2, \ldots, j_k < t$ *such that* $g_U(X_{j_1}, \ldots, X_{j_k}) = (Y_{j_1}, \ldots, Y_{j_k})$ **then**
        Advance the game:
        $U \leftarrow U \cup \{((X_{j_1}, \ldots, X_{j_k}), (Y_{j_1}, \ldots, Y_{j_k}))\}$.
        $k \leftarrow k + 1$.
        $L \leftarrow \emptyset$.
        $m \leftarrow 1$.
        $t' \leftarrow t - 1$.
    **end**
    Predict

$$\hat{Y}_t = \operatorname*{argmax}_y \Pr\left[w(\mathcal{H}^{g_U}_{L \cup (X_t, 1-y)}, X_{[t, t(m)+t']}) \leq \frac{1}{2} w(\mathcal{H}^{g_U}_L, X_{[t, t(m)+t']}) \,\middle|\, X_{\leq t}\right]$$

    **if** $Y_t \neq \hat{Y}_t$ **then**
        $L \leftarrow L \cup \{(X_t, Y_t)\}$.
    **end**
    **if** $t \geq \frac{m(m+1)}{2} + t'$. **then**
        $m \leftarrow m + 1$.
    **end**
**end**

---

The following lemma from the work of Bousquet et al. [2021] holds:

**Lemma 14** (Bousquet et al. [2021]). *For any process $\{(X_i, Y_i)_{i \in \mathbb{N}}\} \in R(\mathcal{H})$, there exists $t_0$, such that for all $t \geq t_0$, algorithm 1 will not update $k$ and $U$ and for all $j_1, j_2, \ldots, j_k$, the winning strategy $g_U$ satisfies*

$$g_U(X_{j_1}, \ldots, X_{j_k}) \neq (Y_{j_1}, \ldots, Y_{j_k}).$$

*Proof.* By the definition of the winning strategy, it leads to a winning condition for the player $P_L$. By the definition of $P_L$'s winning condition, we know that there exists a $k$ such that $\mathcal{H}_{X_1, g_1, \ldots, X_k, g_k} = \emptyset$, which means for all $j_1, j_2, \ldots, j_k, g_U(X_{j_1}, \ldots, X_{j_k}) \neq (Y_{j_1}, \ldots, Y_{j_k})$. That finishes the proof. $\quad\square$

This lemma shows that if the concept class $\mathcal{H}$ has no infinite VCL tree, for a sufficiently large $t_0$, the VCL game will stop advancing after round $t_0$. Once the game stops advancing, we are effectively just bounding the number of mistakes by a predictor based on a partial concept class of finite VC dimension. This result is interesting in its own right, we extract this portion of the algorithm into a separate subroutine, which is stated as Algorithm 2 in AppendixA.1, for which we prove the following result.

**Lemma 15.** *For any process $\mathbb{X}$ and $\mathcal{H}$ be a partial concept class on $\mathbb{X}$ with $\mathrm{VC}(\mathcal{H}) = d < \infty$. The subroutine (Algorithm 2 in AppendixA.1) only makes $o(T)$ mistakes almost surely as $T \to \infty$.*

For brevity, we put the proof of this lemma in the appendix. The intuition behind the proof is that every mistake decreases the weight by at least half with more than half probability, so the number of mistakes is $o(T)$.

Combining the lemmas above, for a concept class $\mathcal{H}$ with no infinite VCL tree, for any realizable sequence, Algorithm 1 satisfies $\limsup_{n\to\infty} \frac{1}{n} \sum_{t=1}^{n} \mathbb{I}\left[Y_t \neq \hat{h}_{t-1}(X_t)\right] = 0$ a.s. Because the winning strategy only updates finite times, the long-run average number of mistakes is dominated by the number of mistakes made by the subroutine, which is $o(n)$.

To prove the necessity, we show that for every concept class $\mathcal{H}$ with an infinite VCL tree, there exists at least one process that does not admit universal online learning under $\mathcal{H}$. Formally,

**Theorem 16.** *For every concept class $\mathcal{H}$ with infinite VCL tree, there exists a process $\mathbb{X}$, such that $\mathbb{X}$ does not admit universal consistent online learning.*

We need the following definition and results from Bousquet et al. [2023] to define the sequence.

**Notation 17** (Bousquet et al. [2023]). *For any $\mathbf{u} \in (\{0, 1\})^*$, let index$(\mathbf{u}) \in \mathbb{N}$ denote the index of $\mathbf{u}$ in the lexicographic ordering of $(\{0, 1\})^*$.*

**Definition 18** (Bousquet et al. [2023]). *Let $\mathcal{X}$ be a set and $\mathcal{H} \subseteq \{0, 1\}^{\mathcal{X}}$ be a hypothesis class, and let*

$$T = \{x_{\mathbf{u}} \in \mathcal{X} : \mathbf{u} \in (\{0, 1\})^*\}$$

*be an infinite VCL tree that is shattered by $\mathcal{H}$. This implies the existence of a collection*

$$\mathcal{H}_T = \{h_{\mathbf{u}} \in \mathcal{H} : \mathbf{u} \in (\{0, 1\})^*\}$$

*of consistent functions.*

*We say such a collection is **indifferent** if for every $\mathbf{v}, \mathbf{u}, \mathbf{w} \in (\{0, 1\})^*$, if index$(\mathbf{v}) <$ index$(\mathbf{u})$, and $\mathbf{w}$ is a descendant of $\mathbf{u}$ in the tree $T$, then $h_{\mathbf{u}}(x_{\mathbf{v}}) = h_{\mathbf{w}}(x_{\mathbf{v}})$. In other words, the functions for all the descendants of a node that appears after $\mathbf{v}$ agree on $\mathbf{v}$.*

*We say that $T$ is **indifferent** if it has a set $\mathcal{H}_T$ of consistent functions that are indifferent.*

**Theorem 19** (Bousquet et al. [2023]). *Let $\mathcal{X}$ be a set and $\mathcal{H} \subseteq \{0, 1\}^{\mathcal{X}}$ be a hypothesis class, and let $T$ be an infinite VCL tree that is shattered by $\mathcal{H}$. Then there exists an infinite VCL tree $T'$ that is shattered by $\mathcal{H}$ that is indifferent.*

Here is the proof sketch of Theorem 16

*Proof Sketch.* First, we can modify the indifferent infinite VCL tree such that it has the property that the number of elements contained by the $k$-th node in the Breadth-First-Search (BFS) order is $2^{k-1}$. The data process we are choosing is all the data come in the lexical order in each node and the BFS order among different nodes. Then we take a random walk on this tree to choose the true label for each instance. The instances in the node visited by the random walk will be labeled by the label on the edge adjacent to it in the path. The instances in the node that is off-branch will be labeled by the label decided by its descendants. (We can do this as the tree is indifferent.) Thus, when reaching a node on the path, no matter what the algorithm predicts, it makes mistakes with probability $\frac{1}{2}$. Thus, it makes a quarter mistake in expectation. Then by Fatou's lemma, for each learning algorithm, we get a realizable process such that the algorithm does not make a sublinear loss almost surely. $\square$

We finish the proof of Theorem 9 here. We then focus on the existence of the optimistically universal online learner when all processes admit universal online learning.

### 4.1 Optimistically Universal Online Learning Rule

In this part, we show that the condition whether $\mathcal{H}$ has an infinite Littlestone tree is the sufficient and necessary condition for the existence of an optimistically universal online learning rule, when all processes admit universal online learning. This is formally stated as theorem 10. The sufficiency part of theorem 10 is proved in Bousquet et al. [2021] as the following lemma:

**Lemma 20** (Bousquet et al. [2021] Theorem 3.1, the first bullet). *If $\mathcal{H}$ does not have an infinite Littlestone tree, then there is a strategy for the learner that makes only finitely many mistakes against any adversary.*

Notice that the online learning algorithm derived from the winning strategy of the learner only makes finite mistakes against any adversary, so for every realizable data process $(\mathbb{X}, \mathbb{Y})$, this online learning algorithm also only makes finite mistakes, which means the long-run average mistake bound goes to 0. Thus, this is an optimistically universal online learning rule, and the concept class $\mathcal{H}$ which does not have an infinite Littlestone tree is optimistically universally online learnable.

The necessity is restated as the following theorem:

**Theorem 21.** *For any concept class $\mathcal{H}$ with an infinite Littlestone tree, for any online learning algorithm $\mathcal{A}$, there exists a process $\mathbb{X}$ that makes $\mathcal{A}$ have an average loss greater than a half with non-zero probability.*

*Proof Sketch.* We can take a random walk on the infinite Littlestone tree to generate the target function. Thus, no matter what the algorithm predicts, it makes a mistake with a probability of more than half. Then we can use Fatou's lemma to get a lower bound of the expected average loss of the learning algorithm among all random processes and that means for every algorithm there exists a process that makes its average loss more than a half with probability more than zero. $\square$

## 5 Concept Classes with an Infinite VCL Tree

In this section, we discuss the necessary and sufficient conditions for a process to admit universal online learning under the concept class $\mathcal{H}$ with an infinite VCL tree. Theorem 11 states the answer formally. To prove this theorem, we first prove sufficiency (Lemma 22) and then necessity (Lemma 23).

**Lemma 22.** *If a process $\mathbb{X}$ satisfies condition A, it admits universally consistent online learning under concept class $\mathcal{H}$.*

*Proof Sketch.* To prove this lemma, we use the weighted majority algorithm with non-uniform initial weights on the experts defined in condition A. The initial weight of expert $i$ is $\frac{1}{i(i+1)}$, where the index $i$ is the index defined in condition A as well. $\square$

**Lemma 23.** *If a process $\mathbb{X}$ admits universally consistent online learning under concept class $\mathcal{H}$, it satisfies condition A.*

*Proof Sketch.* In order to prove this lemma, we need to show the following statement holds:

For a given concept class $\mathcal{H}$, and a data process $\mathbb{X}$, if there is a learning algorithm $\mathcal{A}$ that is strongly universal consistent under $\mathbb{X}$ and $\mathcal{H}$, then we have a set of experts $E = \{e_1, e_2, \dots\}$, there is a sequence $\{i_n\}$ with $\log i_n = o(n)$, such that for any realizable sequence $(\mathbb{X}, \mathbb{Y})$, for any $n \in \mathbb{N}$, there is an expert $e_i$ with $i \leq i_n$ such that $Y_t = e_i(X_t)$ for every $t \leq n$.

We modify the construction from the work of Ben-David et al. [2009] to build the experts. The experts are based on the set of the indexes of the rounds when the algorithm $\mathcal{A}$ makes mistakes, so there is a one-on-one map from the set of the indexes of the mistakes to the experts. Then we can index the experts based on the set of the indexes of mistakes to show the existence of such a sequence. $\square$

Then we can get the theorem for optimistically universal online learnability, which is theorem 12. Because the proof of lemma 23 and 22 works for any process, we can prove Theorem 12 by reusing the proof of Theorem 11.

## 6 Agnostic Case

In this section, we extend the online learning algorithm for realizable cases to an online learning algorithm for agnostic cases. The basic idea follows the idea of Ben-David et al. [2009]. In other

words, we build an expert for each realizable process $(\mathbb{X}, \mathbb{Y})$. Then we run the learning with experts' advice algorithm on those experts and get a low regret learning algorithm.

**Theorem 24.** *The following two statements are equivalent:*

- *There is an online learning rule that is strongly universally consistent under $\mathbb{X}$ and $\mathcal{H}$ for the realizable case.*

- *There is an online learning rule that is strongly universally consistent under $\mathbb{X}$ and $\mathcal{H}$ for the agnostic case.*

*Proof Sketch.* To prove this lemma, we first build the experts $e_i$ based on the learning algorithm for the realizable case by using the construction from lemma 23. We then use the learning on experts' advice algorithm called *Squint* from Koolen and van Erven [2015], with non-uniform initial weights $\frac{1}{i(i+1)}$ for each $e_i$ to get sublinear regret. Thus, we can extend the learning algorithm for realizable cases to a learning algorithm for agnostic cases no matter how the algorithm operates.

An online learning algorithm for the agnostic case is also an online learning algorithm for the realizable case, by taking $\mathbb{Y}^* = \mathbb{Y}$, the regret becomes the number of mistakes. Thus, the two statements are equivalent. $\square$

Theorem 24 implies that all the characterizations for the realizable case are also characterizations for the agnostic case. We formally state the following theorems:

**Theorem 25.** *For the agnostic case and any concept class $\mathcal{H}$ with no infinite VCL tree, any process $\mathbb{X}$ admits strongly universal online learning under $\mathcal{H}$. However, only the concept class with no infinite Littlestone tree is optimistically universally online learnable.*

For the concept class $\mathcal{H}$ with infinite VCL tree:

**Theorem 26.** *For the agnostic case, a data process $\mathbb{X}$ admits strongly universal online learning under concept class $\mathcal{H}$ with infinite VCL tree if and only if it satisfies condition A. However, a concept class $\mathcal{H}$ with infinite VCL tree is optimistically universally online learnable if and only if it satisfies condition B.*

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

# A  Omitted Proofs

## A.1  Proof of lemma 15

In order to help the reader understand, we provide the subroutine here again for reference.

---

**Algorithm 2:** Subroutine for learning a partial concept class $\mathcal{H}$ with VC dimension $d$ on the data process $\mathbb{X}$.

---

$L \leftarrow \emptyset$.
$m \leftarrow 1$.
**for** $t = 1, 2, 3, \ldots$ **do**

    Predict

$$\hat{Y}_t = \underset{y}{\operatorname{argmax}} \Pr \left[ w(\mathcal{H}_{L \cup (X_t, 1-y)}^{g_U}, X_{[t, t(m)]}) \leq \frac{1}{2} w(\mathcal{H}_L^{g_U}, X_{[t, t(m)]}) \,\middle|\, X_{\leq t} \right]$$

    **if** $Y_t \neq \hat{Y}_t$ **then**
    |   $L \leftarrow L \cup \{(X_t, Y_t)\}$.
    **end**

    **if** $t \geq \frac{m(m+1)}{2}$. **then**
    |   $m \leftarrow m + 1$.
    **end**

**end**

---

*Proof.* In this proof, we assume that for the partial concept class $\mathcal{H}$ with VC dimension $\leq d$, $\{(X_i, Y_i)\}_{i \in \mathbb{N}}$ is realizable. As we defined, the weight function, $w(\mathcal{H}', X_{\leq T}) = |\{S : S \subseteq \{X_i\}_{i \leq T}$ such that $S$ shattered by $\mathcal{H}'\}|$, which is the number of the subsequences of the sequence $X_{\leq T}$ that can be shattered by the partial concept class $\mathcal{H}'$.

Consider the $k$-th batch, consisting of $W_k = \{X_{\frac{k(k-1)}{2}+1}, \cdots, X_{\frac{k(k+1)}{2}}\}$. Let

$$Z_k = \sum_{t=\frac{k(k-1)}{2}+1}^{\frac{k(k+1)}{2}} \mathbb{I}\left[\hat{Y}_t \neq Y_t\right],$$

and

$$V_k = Z_k - \mathbb{E}\left[Z_k \,\middle|\, X_{\leq \frac{k(k-1)}{2}}\right].$$

Notice that

$$\mathbb{E}\left[V_k \,\middle|\, X_{\leq \frac{k(k-1)}{2}}\right]$$
$$= \mathbb{E}\left[Z_k - \mathbb{E}\left[Z_k \,\middle|\, X_{\leq \frac{k(k-1)}{2}}\right] \,\middle|\, X_{\leq \frac{k(k-1)}{2}}\right]$$
$$= \mathbb{E}\left[Z_k \,\middle|\, X_{\leq \frac{k(k-1)}{2}}\right] - \mathbb{E}\left[Z_k \,\middle|\, X_{\leq \frac{k(k-1)}{2}}\right] = 0. (a.s.)$$

Thus, the sequence $V_k$ is a martingale difference sequence with respect to the block sequence, $W_1, W_2, \cdots$. By the definition of $V_k$, we also have $-k \leq V_k \leq k$. Then by Azuma's Inequality, with probability $1 - \frac{1}{K^2}$, we have

$$\sum_{k=1}^{K} Z_k \leq \sum_{k=1}^{K} \mathbb{E}\left[Z_k \,\middle|\, X_{\leq \frac{k(k-1)}{2}}\right] + \sqrt{-\log\left(\frac{1}{K^2}\right) \cdot 2 \cdot \left(\sum_{k=1}^{K} k^2\right)}$$

$$\leq \sum_{k=1}^{K} \mathbb{E}\left[Z_k \,\middle|\, X_{\leq \frac{k(k-1)}{2}}\right] + \sqrt{4K^3 \log K}.$$

Then we need to get an upper bound for $\mathbb{E}\left[Z_k \,\middle|\, X_{\le \frac{k(k-1)}{2}}\right]$. According to the prediction rule, every time we make a mistake, we have

$$\Pr\left[w(\mathcal{H}^{g_U}_{L \cup (X_t, Y_t)}, X_{[t, \frac{k(k+1)}{2}]}) \le \frac{1}{2} w(\mathcal{H}^{g_U}_L, X_{[t, \frac{k(k+1)}{2}]}) \,\middle|\, X_{\le t}\right] \ge \frac{1}{2}. \tag{3}$$

Due to the linearity of expectation, for every $k$,

$$\mathbb{E}\left[\sum_{t=\frac{k(k-1)}{2}}^{\frac{k(k+1)}{2}} \mathbb{I}\left[\hat{Y}_t \ne Y_t\right] \,\middle|\, X_{\le \frac{k(k-1)}{2}}\right]$$

$$= \mathbb{E}\left[\sum_{t=\frac{k(k-1)}{2}}^{\frac{k(k+1)}{2}} \mathbb{I}\left[\hat{Y}_t \ne Y_t\right] \mathbb{I}\left[w(\mathcal{H}_{L_t}, X_{[t+1, \frac{k(k+1)}{2}]}) \le \frac{1}{2} w(\mathcal{H}_{L_{t-1}}, X_{[t, \frac{k(k+1)}{2}]})\right] \,\middle|\, X_{\le \frac{k(k-1)}{2}}\right]$$

$$+ \mathbb{E}\left[\sum_{t=\frac{k(k-1)}{2}}^{\frac{k(k+1)}{2}} \mathbb{I}\left[\hat{Y}_t \ne Y_t\right] \mathbb{I}\left[w(\mathcal{H}_{L_t}, X_{[t+1, \frac{k(k+1)}{2}]}) > \frac{1}{2} w(\mathcal{H}_{L_{t-1}}, X_{[t, \frac{k(k+1)}{2}]})\right] \,\middle|\, X_{\le \frac{k(k-1)}{2}}\right].$$

Here $L_t = \{(X_i, Y_i)\}$, where $i \le t$ and the algorithm makes a mistake at round $i$.

Notice the first part is the expected number of mistakes, each of which decreases the weight by half. For every realization of $X_{[\frac{k(k-1)}{2}, \frac{k(k+1)}{2}]}$, $x_{[\frac{k(k-1)}{2}, \frac{k(k+1)}{2}]}$, let

$$u(k) = \sum_{i=\frac{k(k-1)}{2}}^{\frac{k(k+1)}{2}} \mathbb{I}\left[\hat{Y}_t \ne Y_t\right] \mathbb{I}\left[w(\mathcal{H}_{L_t}, x_{[t+1, \frac{k(k+1)}{2}]}) \le \frac{1}{2} w(\mathcal{H}_{L_{t-1}}, x_{[t, \frac{k(k+1)}{2}]})\right].$$

By the definition of the weight function and the fact that $\mathrm{VC}(\mathcal{H}) = d$, $w(\mathcal{H}_{L_{\frac{k(k-1)}{2}}}, x_{[\frac{k(k-1)}{2}, \frac{k(k+1)}{2}]}) \le k^d$. Consider the last round $t \le \frac{k(k+1)}{2}$ that $\hat{Y}_t \ne Y_t$, we have $w(\mathcal{H}_{L_{t-1}, x_{[t, \frac{k(k+1)}{2}]}}) \ge 1$, as the set $\{x_t\}$ must be shattered. Thus, we have $2^{u(k)-1} w(\mathcal{H}_{L_{t-1}}, x_{[t, \frac{k(k+1)}{2}]}) \le w(\mathcal{H}, x_{[\frac{k(k-1)}{2}, \frac{k(k+1)}{2}]})$. Therefore, $u(k) \le d \log k + 1$, for every realization, $x_{[\frac{k(k-1)}{2}, \frac{k(k+1)}{2}]}$. Thus,

$$\mathbb{E}\left[\sum_{t=\frac{k(k-1)}{2}}^{\frac{k(k+1)}{2}} \mathbb{I}\left[\hat{Y}_t \ne Y_t\right] \mathbb{I}\left[w(\mathcal{H}_{L_t}, X_{[t+1, \frac{k(k+1)}{2}]}) \le \frac{1}{2} w(\mathcal{H}_{L_{t-1}}, X_{[t, \frac{k(k+1)}{2}]})\right] \,\middle|\, X_{\le \frac{k(k-1)}{2}}\right] \le 2d \log k. \tag{4}$$

Then consider the second part, we have

$$\mathbb{E}\left[\sum_{t=\frac{k(k-1)}{2}}^{\frac{k(k+1)}{2}} \mathbb{I}\left[\hat{Y}_t \ne Y_t\right] \mathbb{I}\left[w(\mathcal{H}_{L_t}, X_{[t+1, \frac{k(k+1)}{2}]}) > \frac{1}{2} w(\mathcal{H}_{L_{t-1}}, X_{[t, \frac{k(k+1)}{2}]})\right] \,\middle|\, X_{\le \frac{k(k-1)}{2}}\right]$$

$$= \mathbb{E}\left[\mathbb{E}\left[\sum_{t=\frac{k(k-1)}{2}}^{\frac{k(k+1)}{2}} \mathbb{I}\left[\hat{Y}_t \ne Y_t\right] \mathbb{I}\left[w(\mathcal{H}_{L_t}, X_{[t+1, \frac{k(k+1)}{2}]}) > \frac{1}{2} w(\mathcal{H}_{L_{t-1}}, X_{[t, \frac{k(k+1)}{2}]})\right] \,\middle|\, X_{\le t}\right] \,\middle|\, X_{\le \frac{k(k-1)}{2}}\right]$$

$$= \mathbb{E}\left[\sum_{t=\frac{k(k-1)}{2}}^{\frac{k(k+1)}{2}} \mathbb{I}\left[\hat{Y}_t \ne Y_t\right] \mathbb{E}\left[\mathbb{I}\left[w(\mathcal{H}_{L_t}, X_{[t+1, \frac{k(k+1)}{2}]}) > \frac{1}{2} w(\mathcal{H}_{L_{t-1}}, X_{[t, \frac{k(k+1)}{2}]})\right] \,\middle|\, X_{\le t}\right] \,\middle|\, X_{\le \frac{k(k-1)}{2}}\right]$$

This is because $\hat{Y}_t$ and $Y_t$ only depend on $X_{\leq t}$. Due to the equation 3, we have

$$\mathbb{I}\left[\hat{Y}_t \neq Y_t\right]\mathbb{E}\left[\mathbb{I}\left[w(\mathcal{H}_{L_t}, X_{[t+1, \frac{k(k+1)}{2}]}) > \frac{1}{2}w(\mathcal{H}_{L_{t-1}}, X_{[t, \frac{k(k+1)}{2}]})\right]\middle| X_{\leq t}\right]$$

$$= \mathbb{I}\left[\hat{Y}_t \neq Y_t\right]\Pr\left[w(\mathcal{H}_{L_t}, X_{[t+1, \frac{k(k+1)}{2}]}) > \frac{1}{2}w(\mathcal{H}_{L_{t-1}}, X_{[t, \frac{k(k+1)}{2}]})\middle| X_{\leq t}\right]$$

$$\leq \frac{1}{2}\mathbb{I}\left[\hat{Y}_t \neq Y_t\right].$$

Thus,

$$\mathbb{E}\left[\sum_{t=\frac{k(k-1)}{2}}^{\frac{k(k+1)}{2}} \mathbb{I}\left[\hat{Y}_t \neq Y_t\right]\mathbb{I}\left[w(\mathcal{H}_{L_t}, X_{[t+1, \frac{k(k+1)}{2}]}) > \frac{1}{2}w(\mathcal{H}_{L_{t-1}}, X_{[t, \frac{k(k+1)}{2}]})\right]\middle| X_{\leq \frac{k(k-1)}{2}}\right] \quad (5)$$

$$\leq \frac{1}{2}\mathbb{E}\left[\sum_{t=\frac{k(k-1)}{2}}^{\frac{k(k+1)}{2}} \mathbb{I}\left[\hat{Y}_t \neq Y_t\right]\middle| X_{\leq \frac{k(k-1)}{2}}\right]$$

Combining these two inequalities (4 and 5), we have

$$\mathbb{E}\left[\sum_{t=\frac{k(k-1)}{2}}^{\frac{k(k+1)}{2}} \mathbb{I}\left[\hat{Y}_t \neq Y_t\right]\middle| X_{\leq \frac{k(k-1)}{2}}\right] \leq 2d\log k + \frac{1}{2}\mathbb{E}\left[\sum_{t=\frac{k(k-1)}{2}}^{\frac{k(k+1)}{2}} \mathbb{I}\left[\hat{Y}_t \neq Y_t\right]\middle| X_{\leq \frac{k(k-1)}{2}}\right]. \quad (6)$$

Thus, for any $k$, we have

$$\mathbb{E}\left[\sum_{t=\frac{k(k-1)}{2}}^{\frac{k(k+1)}{2}} \mathbb{I}\left[\hat{Y}_t \neq Y_t\right]\middle| X_{\leq \frac{k(k-1)}{2}}\right] \leq 4d\log k. \quad (7)$$

According to the inequality 7 for every $k$, $\mathbb{E}\left[Z_k\middle| X_{\leq \frac{k(k-1)}{2}}\right] \leq 4d\log k$. Thus, with probability at least $1 - \frac{1}{K^2}$,

$$\sum_{k=1}^{K} Z_k \leq \sum_{k=1}^{K} 4d\log k + \sqrt{4K^3\log K} \leq 4dK\log K + \sqrt{4K^3\log K}.$$

By the definition of $Z_k$, with probability at least $1 - \frac{1}{K^2}$,

$$\sum_{t=1}^{\frac{K(K+1)}{2}} \mathbb{I}\left[\hat{Y}_t \neq Y_t\right] \leq 4dK\log K + \sqrt{4K^3\log K} \leq (4d+2)\sqrt{K^3\log K}. \quad (8)$$

Let $T_K = \frac{K(K+1)}{2}$ be the number of instances in the sequence, with probability at least $1 - \frac{1}{K^2}$

$$\sum_{t=1}^{T_K} \mathbb{I}\left[\hat{Y}_t \neq Y_t\right] \leq (4d+2)T_K^{\frac{3}{4}}\sqrt{\frac{1}{2}\log T_K}. \quad (9)$$

Define the event $E_K$ as the event that in the sequence $X_{\leq T_K}$, $\sum_{t=1}^{T_K} \mathbb{I}\left[\hat{Y}_t \neq Y_t\right] > (4d + 2)T_K^{\frac{3}{4}}\sqrt{\frac{1}{2}\log T_K}$. Then we know $\Pr[E_K] \leq \frac{1}{K^2}$. Notice the fact that for any $K \in \mathbb{N}$, $\sum_{k=1}^{K} \frac{1}{k^2} \leq \frac{\pi^2}{6}$. By Borel-Cantelli lemma, we know that for any $T_K = \frac{K(K+1)}{2}$ large enough, $\sum_{t=1}^{T_K} \mathbb{I}\left[\hat{Y}_t \neq Y_t\right] \leq (4d + 2)T_K^{\frac{3}{4}}\sqrt{\frac{1}{2}\log T_K}$ happens with probability 1.

Then for any large enough $T$, we have $T_K \leq T \leq T_{K+1} \leq 2T$. Thus, with probability 1,

$$\sum_{t=1}^{T} \mathbb{I}\left[\hat{Y}_t \neq Y_t\right] \leq (4d+2)T_{K+1}^{\frac{3}{4}}\sqrt{\frac{1}{2}\log T_{K+1}} \quad (10)$$

$$\leq (4d+2)(2T)^{\frac{3}{4}}\sqrt{\frac{1}{2}\log 2T}. \quad (11)$$

Therefore, for any large enough $T$ and a universal constant $c$, with probability 1,

$$\sum_{t=1}^{T} \mathbb{I}\left[\hat{Y}_t \neq Y_t\right] \leq cT^{\frac{3}{4}}\sqrt{\log T}. \tag{12}$$

Notice that $cT^{\frac{3}{4}}\sqrt{\log T}$ is $o(T)$. That finishes the proof.

$\square$

## A.2  Proof of Theorem 16

*Proof.* In this proof, we first modify the chosen indifferent VCL tree, such that the number of elements in each node is increasing exponentially. In other words, we hope the $k$-th node in the Breadth-First-Search (BFS) order contains $2^{k-1}$ elements. We may reach this target by recursively modifying the chosen VCL tree. Starting from the root of the tree, for each node that does not satisfy our requirement, we promote one of its descendants to replace that node, such that the number of elements in that node is large enough.

Then we define the data process as follows: For the modified VCL tree, we define the sequence $\{X_i\}_{i \in \mathbb{N}}$ as $X_{2^{k-1}+j} = X_{jk}$, where $X_{jk}$ is the $j$-th element in the $k$-th node in the BFS order.

Next, we define the target function, in other words, choose the label $Y_t$ for each $X_t$. First, we take a random walk in the modified VCL tree. Then for the elements in the node on the path we visited (in-branch node), we let $Y_t$ be the label given by the edge adjacent to that node. Then, we need to decide the label of those elements in the node not on the path we visited (off-branch nodes). For any off-branch node, we can pick an in-branch node after it in BFS order, as the tree is indifferent, all descendants of the in-branch node agree on the label of the elements in the off-branch node. Thus, we can let the label of the elements in the off-branch node be the label decided by the descendant of that in-branch node. So, every element in the node that is visited by the random walk still may be wrong with probability $\frac{1}{2}$, when the algorithm sees it the first time. Also, the number of elements that come before $k$-th node in the BFS order, $\sum_{i=0}^{k-2} 2^i = 2^{k-1} - 1$, is roughly the same as the number of elements in the $k$-th node, $2^{k-1}$. Thus at the $d$-th layer of the modified tree, if the random walk reaches the node $K_d$, for that node, we have

$$\mathbb{E}\left[\frac{1}{n_{K_d}}\sum_{t=1}^{n_{K_d}} \mathbb{I}\left[h_{t-1}(X_t) \neq Y_t\right] \,\middle|\, K_d\right] \geq \frac{1}{4}. \tag{13}$$

Here $n_{K_d}$ is the number of elements in the process when we reach the $K_d$-th node. This inequality holds for all $d$.

Then notice that by taking an expectation on the expected mistakes for every deterministic sequence, we get an expectation of the number of mistakes for this random process. Then we can pick the sub-sequence, which only contains $n_{K_d} = 2^{K_d} - 1$ elements, and this decreases the ratio of mistakes (the third line in the following computations). This is because we can only make mistakes when the elements are in the $K_d$-th node and any other $n$ will have a smaller ratio of mistakes than $n_{K_d}$. Then notice that the ratio of mistakes is always smaller than or equal to 1, we can use the reversed Fatou's lemma and inequality 13 to get the final result (the fourth line and the last line in the following computations).

$$\mathbb{E}\left[\mathbb{E}\left[\limsup_{n\to\infty}\frac{1}{n}\sum_{t=1}^{n}\mathbb{I}\left[h_{t-1}(X_t)\neq Y_t\right]\middle|(\mathbb{X},\mathbb{Y})\right]\right]$$

$$=\mathbb{E}\left[\limsup_{n\to\infty}\frac{1}{n}\sum_{t=1}^{n}\mathbb{I}\left[h_{t-1}(X_t)\neq Y_t\right]\right]$$

$$\geq\mathbb{E}\left[\limsup_{d\to\infty}\frac{1}{n_{K_d}}\sum_{t=1}^{n_{K_d}}\mathbb{I}\left[h_{t-1}(X_t)\neq Y_t\right]\right]$$

$$\geq\mathbb{E}\left[\limsup_{d\to\infty}\mathbb{E}\left[\frac{1}{n_{K_d}}\sum_{t=1}^{n_{K_d}}\mathbb{I}\left[h_{t-1}(X_t)\neq Y_t\right]\middle|n_{K_d}\right]\right]$$

$$\geq\frac{1}{4}.$$

Thus, there exists a deterministic sequence $(\mathbb{X},\mathbb{Y})$ such that it does not make sublinear expected mistakes. $\square$

### A.3 Proof of Theorem 21

*Proof.* As $\mathcal{H}$ has an infinite Littlestone tree, we can take a random walk on this tree, then for every step $t$, take the label of the node as $X_t$, and no matter what the learning algorithm predicts, uniformly randomly choose $Y_t$. Thus, we have $\mathbb{E}\left[\mathbb{I}\left[h_{t-1}(X_t)\neq Y_t\right]\right]\geq\frac{1}{2}$ for every $t$. We get

$$\limsup_{n\to\infty}\mathbb{E}_{(\mathbb{X},\mathbb{Y})}\left[\frac{1}{n}\sum_{t=1}^{n}\mathbb{I}\left[h_{t-1}(X_t)\neq Y_t\right]\right]\geq\frac{1}{2}. \tag{14}$$

According to Fatou's lemma, notice the ratio of mistakes is smaller than or equal to 1, so we have the following inequality,

$$\mathbb{E}\left[\limsup_{n\to\infty}\frac{1}{n}\sum_{t=1}^{n}\mathbb{I}\left[h_{t-1}(X_t)\neq Y_t\right]\right]\geq\frac{1}{2}. \tag{15}$$

Thus, for each learning algorithm, there exists a data sequence $\{(X_t,Y_t)\}_{t\in\mathbb{N}}$ such that equation 15 holds. That finishes the proof. $\square$

### A.4 Proof of Lemma 22

For the completeness, we provide the weighted majority algorithm here:

---

**Algorithm 3:** The Weighted Majority Algorithm with Non-uniform Initial Weights

---

For expert $e_i$, assign $w_i^0=\frac{1}{i(i+1)}$ as its initial weight.

**for** $t = 1,2,\ldots$ **do**

$\quad$ Predict $y_t=\mathbb{I}\left[\sum_i\frac{w_i^{t-1}}{\sum_i w_i^{t-1}}e_i(X_t)\geq\frac{1}{2}\right]$.

$\quad$ Update $w_i^t=\left(\frac{1}{2}\right)^{\mathbb{I}[e_i(X_t)\neq Y_t]}w_i^{t-1}$.

**end**

---

By using this algorithm, we provide the proof of the lemma 22.

*Proof.* In order to prove this lemma, we use the weighted majority algorithm 3 with initial weight $w_i^0=\frac{1}{i(i+1)}$ for each expert $i$. We set MB $=\sum_{t=1}^{n}\mathbb{I}\left[h_{t-1}(X_t)\neq Y_t\right]$, which is the number of mistakes the algorithm made during $n$ rounds. We also set $m_i=\sum_{t=1}^{n}\mathbb{I}\left[e_i(X_t)\neq Y_t\right]$. Next, compute the total weight of all experts after $n$ rounds of the algorithm, $W^n$. Notice

that if the algorithm makes a mistake at round $n$, there must be a majority of the experts making a mistake at round $n$, so $W^{n-1} - W^n \geq \frac{1}{2} \cdot \frac{1}{2} W^{n-1}$, which means $W^n \leq \frac{3}{4} W^{n-1}$. Thus, we have $W^n \leq \left(\frac{3}{4}\right)^{\text{MB}} W^0 \leq \left(\frac{3}{4}\right)^{\text{MB}}$. Notice that $W^n \geq w_i^n$ for all $i$, so it holds for $\operatorname{argmin}_{i \leq i_n} \sum_{t=1}^{n} \mathbb{I}\left[e_i(X_t) \neq Y_t\right]$. We have the following inequality for all $n$,

$$\sum_{t=1}^{n} \mathbb{I}\left[h_{t-1}(X_t) \neq Y_t\right] \leq 3 \min_{i \leq i_n} \sum_{t=1}^{n} \mathbb{I}\left[e_i(X_t) \neq Y_t\right] + \log\left(\frac{1}{w_i^0}\right) \tag{16}$$

$$\leq 3 \min_{i \leq i_n} \sum_{t=1}^{n} \mathbb{I}\left[e_i(X_t) \neq Y_t\right] + 2 \log i_n. \tag{17}$$

Here 3 comes from the fact that $\frac{\log 2}{\log(\frac{4}{3})} \leq 3$. Therefore, for a fixed process $\mathbb{X}$, target function $h^*$ and a fixed sequence, $\{i_n\}$, we have

$$\mathbb{E}\left[\limsup_{n \to \infty} \frac{1}{n} \sum_{t=1}^{n} \mathbb{I}\left[h_{t-1}(X_t) \neq Y_t\right]\right] \leq \mathbb{E}\left[\limsup_{n \to \infty} \frac{1}{n}\left(3 \min_{i \leq i_n} \sum_{t=1}^{n} \mathbb{I}\left[e_i(X_t) \neq Y_t\right] + 2 \log i_n\right)\right]. \tag{18}$$

Then by the condition A, we know the right-hand side of the inequality is $0$.

Notice that $\limsup_{n \to \infty} \frac{1}{n} \sum_{t=1}^{n} \mathbb{I}\left[h_{t-1}(X_t) \neq Y_t\right]$ is a non-negative random variable, so we have

$$\mathbb{E}\left[\limsup_{n \to \infty} \frac{1}{n} \sum_{t=1}^{n} \mathbb{I}\left[h_{t-1}(X_t) \neq Y_t\right]\right] = 0. \tag{19}$$

Therefore, $\limsup_{n \to \infty} \frac{1}{n} \sum_{t=1}^{n} \mathbb{I}\left[h_{t-1}(X_t) \neq Y_t\right] = 0$ almost surely. $\qquad \square$

### A.5 Proof of Lemma 23

---
**Algorithm 4:** Expert $J$

---
**Input:** set $J$
**for** $t = 1,2,\ldots$ **do**
    recieve $X_t$.
    compute $\tilde{y}_t = f_t^{\mathcal{A}}(X_{<t}, \hat{Y}_{<t}, X_t)$.[a]
    **if** $t \in J$ **then**
        | predict $\hat{y}_t = \neg \tilde{y}_t$
    **else**
        | predict $\hat{y}_t = \tilde{y}_t$
    **end**
**end**

---

[a] $\hat{Y}_{<t} = \{\hat{y}_i\}_{<t}$

*Proof.* In this proof, we show how to define a sequence of experts $\{e_1, e_2, \ldots\}$, such that the condition A is satisfied, if $\mathbb{X}$ admits universal online learning. Let an online learning rule $f_t^{\mathcal{A}}$ be the algorithm that can learn all realizable label processes $\mathbb{Y} \in R(\mathcal{H}, \mathbb{X})$. Then we can build the experts by algorithm 4 and represent the experts by the set of the index of mistake rounds. We can define this set as $J$. For example, if the algorithm makes mistakes at round $1, 4, 7$ then the set $J$ for that expert is $\{1, 4, 7\}$. Thus we have a one-on-one map from $J$ to an expert.

First, we show that for every realizable label process $\mathbb{Y} \in R(\mathcal{H}, \mathbb{X})$ and any $n \in \mathbb{N}$, there is an expert $e_i$ such that $e_i(X_t) = Y_t$ for all $t \leq n$. This part of the proof is similar to the proof of lemma 12 in the work of Ben-David et al. [2009]. Consider a subsequence $Y_{\leq n}$ of a realizable label process $\mathbb{Y}$, $j \in J$ if and only if $f_j^{\mathcal{A}}(X_{<j}, \hat{Y}_{<j}, X_j) \neq Y_j$ for all $j \leq n$. Thus, the history $(X_{<t}, \hat{Y}_{<t})$ for all $t \leq n$ are the same as $(X_{<t}, Y_{<t})$, which implies $e_i(X_t) = \hat{Y}_t = Y_t$ for all $t \leq n$. Therefore, for any $n \in \mathbb{N}$, there is a set $J_{i_n}$ containing all $j \leq n$, such that $f_j^{\mathcal{A}}(X_{<j}, \hat{Y}_{<j}, X_j) \neq Y_j$. Then the algorithm 4 with input $J_{i_n}$ creates an expert $e_{i_n}$, such that $e_{i_n}(X_t) = Y_t$ for all $t \leq n$.

Next, we only need to build the index $i_n$ for the set $J_n$ to show that $\log i_n = o(n)$. The index of set $J$ is as follows: For all $J \subseteq \mathbb{N}$, order them by $|J|(\max J)$. (If two sets have the same value, we use $|J|$ as a tie-breaking.) Here $|J|$ is the number of elements in $J$ and $\max J$ is the maximal element in $J$. After that, index $J$'s from 1 following this order.

At last, we show the method mentioned above constructed a set of experts satisfying condition A. we have a set of experts $E = \{e_1, e_2, \dots\}$, there is a sequence $\{i_n\}$ with $\log i_n = o(n)$, such that for any realizable sequence $(\mathbb{X}, \mathbb{Y})$, for any $n \in \mathbb{N}$, there is an expert $e_i$ with $i \leq i_n$ such that $Y_t = e_i(X_t)$ for every $t \leq n$. Therefore, we need to compute the $i_n$ as follows. Assume $|J_{i_n}| \max J_{i_n} = k$, we have

$$i_n \leq |\{J : |J| \max J \leq k\}| = 1 + \sum_{m=1}^{k} |\{J : |J| \leq \frac{k}{m}, \max J = m\}|$$

$$= 1 + \sum_{m=1}^{\sqrt{k}} 2^{m-1} + \sum_{m=\sqrt{k}}^{k} \binom{m-1}{\leq (\frac{k}{m} - 1)} \leq 2^{\sqrt{k}} + \sum_{m=\sqrt{k}}^{k} \left( \frac{em^2}{k} \right)^{\frac{k}{m}} \leq (k+1)e^{\sqrt{k}}.$$

Notice that $k = |J_{i_n}|n$, we have

$$\lim_{n\to\infty} \frac{1}{n} \log i_n \leq \lim_{n\to\infty} \frac{2\sqrt{|J_{i_n}|n}}{n} = 0. \tag{20}$$

Thus, we get the set of experts and the corresponding sequence $\{i_n\}$ satisfying condition A. $\qquad \square$

### A.6 Proof of Theorem 24

*Proof.* In order to prove this theorem, we use the procedure based on learning with experts' advice. First, we build and index the experts, $\{e_1, e_2, \cdots\}$, by using the same method we mentioned in the proof of lemma 23 (A.5), which satisfies Condition A. Then, to use *Squint* algorithm from the work of Koolen and van Erven [2015], we need the initial weight of the experts to be a distribution. We can set the initial weights $\pi_i = \frac{1}{i(i+1)}$ for each $e_i$ and this forms a distribution, as $\pi_i = \frac{1}{i} - \frac{1}{i+1}$, the sum of $\pi_i$ reaches 1 when $i$ goes to infinity.

According to Theorem 3 in the work of Koolen and van Erven [2015], we have the following upper bound for the regret

$$\sum_{t=1}^{T} \mathbb{I}\left[\hat{h}_{t-1}(X_t) \neq Y_t\right] - \sum_{t=1}^{T} \mathbb{I}\left[e_k(X_t) \neq Y_t\right] \leq O\left(\sqrt{V_k \log \frac{\log V_k}{\pi_k}} + \log \frac{1}{\pi_k}\right). \tag{21}$$

Here the $V_k$ is the sum of the square of the difference between the algorithm's mistake and expert $k$'s mistake in each round. In other words, we have

$$V_k = \sum_{i=1}^{T} \left(\mathbb{I}\left[h_{t-1}(X_t) \neq Y_t\right] - \mathbb{I}\left[e_k(X_t) \neq Y_t\right]\right)^2. \tag{22}$$

Notice that $\left(\mathbb{I}\left[h_{t-1}(X_t) \neq Y_t\right] - \mathbb{I}\left[e_k(X_t) \neq Y_t\right]\right)^2$ is either 1 or 0, we have $V_k \leq T$. The regret of this algorithm is upper bounded by:

$$O\left(\sqrt{V_k \log \frac{\log V_k}{\pi_k}} + \log \frac{1}{\pi_k}\right) = O\left(\sqrt{T \log \log T + T \log k} + \log k\right). \tag{23}$$

According to the condition A, we also know

$$\mathbb{E}\left[\limsup_{T\to\infty} \min_{e_i : i \leq i_T} \frac{1}{T} \sum_{t=1}^{T} \mathbb{I}\left[e_i(X_t) \neq Y_t^*\right]\right] = 0. \tag{24}$$

Thus, the regret of this algorithm is

$$\limsup_{T \to \infty} \frac{1}{T} \sum_{t=1}^{T} \left( \mathbb{I} \left[ Y_t \neq \hat{h}_{t-1}(X_t) \right] - \mathbb{I} \left[ Y_t \neq Y_t^* \right] \right)$$

$$= \limsup_{T \to \infty} \frac{1}{T} \sum_{t=1}^{T} \left( \mathbb{I} \left[ Y_t \neq \hat{h}_{t-1}(X_t) \right] - \left| \mathbb{I} \left[ Y_t \neq e_k(X_t) \right] - \mathbb{I} \left[ e_k(X_t) \neq Y_t^* \right] \right| \right)$$

$$\leq \limsup_{T \to \infty} \frac{1}{T} \sum_{t=1}^{T} \left( \mathbb{I} \left[ Y_t \neq \hat{h}_{t-1}(X_t) \right] - \mathbb{I} \left[ Y_t \neq e_k(X_t) \right] + \mathbb{I} \left[ e_k(X_t) \neq Y_t^* \right] \right)$$

$$\leq \limsup_{T \to \infty} \frac{1}{T} \sum_{t=1}^{T} \left( \mathbb{I} \left[ Y_t \neq \hat{h}_{t-1}(X_t) \right] - \mathbb{I} \left[ Y_t \neq e_k(X_t) \right] \right) + \limsup_{T \to \infty} \frac{1}{T} \sum_{t=1}^{T} \mathbb{I} \left[ e_k(X_t) \neq Y_t^* \right]$$

Because $\log i_T = o(T)$ and $\log k < \log i_T$ we have $\log k = o(T)$. Thus, the regret above is $o(T)$. Therefore, we have an algorithm to extend a universally consistent online learning algorithm for realizable cases to a universally consistent online algorithm for agnostic cases.

To prove the necessity, notice that a universally consistent online algorithm for agnostic cases can be used to solve the realizable case and the regret is equal to the number of mistakes. $\square$

