# OpenReview forum: "A Theory of Optimistically Universal Online Learnability for General Concept Classes"
_NeurIPS.cc/2024/Conference — NeurIPS 2024 poster_

### Official Review · Reviewer_jKRy · 2024-07-04

**Soundness:** 3
**Presentation:** 2
**Contribution:** 3
**Rating:** 4
**Confidence:** 2

**Summary:**

This work considers the theory of universal consistency and optimistically universal learning rules in the binary classification case where, instead of considering all measurable target functions, the analysis is parameterized by a specific class $\mathcal{H}$ of binary classifiers. Two types of classes $\mathcal{H}$ are considered:

Strongly universal: for every $\mathcal{H}$-realizable process $\mathbb{X}$ there is a strongly universally consistent online learning algorithm for $\mathcal{H}$ and $\mathbb{X}$.

Optimistically universal: there is an online learning algorithm that is strongly universally consistent for $\mathcal{H}$ and $\mathbb{X}$ for every $\mathcal{H}$-realizable process $\mathbb{X}$.

The main results of this work (as far as I understand) are the following:

$\mathcal{H}$ is strongly universal if and only if it has no infinite VCL tree (Theorem 9).

$\mathcal{H}$ is optimistically universal if and only if it has no infinite Littlestone tree (Theorem 10).

Recall that $\mathcal{H}$ has no infinite Littlestone trees implies $\mathcal{H}$ has no infinite VCL tree.

Further results extends this characterization to the agnostic case.

**Strengths:**

The work considers a variant of universal consistency for online learning in which there is a reference class $\mathcal{H}$ of binary classifiers. A full characterization on universal online learnability is provided.

**Weaknesses:**

The clarity of the presentation is below the NeurIPS standards.

**Questions:**

Please elaborate on the connections between this work and Section 3 of arxiv.org/pdf/2011.04483.

Please clarify the apparent inconsistency in the Theorems 9, 10, and 12.

[The above questions have been addressed in the rebuttal]

**Limitations:**

There is no section on limitations, but the work is purely theoretical with no foreseeable societal impact.

---

> ### Author Rebuttal · Authors · 2024-08-05
>
> We appreciate your helpful comments and we address your questions below.
>
> First, let us re-iterate the definitions of the paper.
> For a given concept class $\mathcal{H}$, a process $\mathbb{X} = (X_1,X_2,\ldots)$ admits a universally consistent online learner if there exists a learning algorithm guaranteeing that, for any realizable classification of the sequence, the learner's number of mistakes grows only sublinearly.  So the property of admitting universal online learning is a property of a concept class $\mathcal{H}$ *and* a stochastic process $\mathbb{X}$.  In contrast, the property of admitting an optimistically universal online learner is a property of a concept class $\mathcal{H}$: namely, $\mathcal{H}$ admits an optimistically universal online learner if there exists an algorithm which is universally consistent (i.e., makes sublinearly many mistakes for realizable classifications) for *every* process  $\mathbb{X}$ which admits a universally consistent online learner.  See the paper (e.g., lines 50-52), and references therein, for further explanation about this terminology and the history of this subject (in the context of prior work, our innovation is incorporating the concept class $\mathcal{H}$ into the theory).
>
> There is no inconsistency in the results.  As we prove (Theorem 9), any class with an infinite VCL tree has the property that the set of processes $\mathbb{X}$ admitting a universally consistent online learner is not the set of *all* processes: that is, only some processes admit a universally consistent online learner.  In contrast, Theorem 9 also establishes that any class that has no infinite VCL tree has the opposite property: that is, *every* process admits a universally consistent online learner.  Note that this is different from the property of admitting an optimistically universal online learner; the latter would require also that there exists a single online learning algorithm which is universally consistent for every such process (in the case of classes having no infinite VCL tree, since Theorem 9 implies every process admits a universally consistent online learner, an optimistically universal learner would need to be universally consistent for *every* process; our Theorem 10 establishes that, among classes with no infinite VCL tree, such an optimistically universal learner exists if and only if $\mathcal{H}$ also has no infinite Littlestone tree).  For classes which have an infinite VCL tree, the property of being an optimistically universal learner only requires the learner to be universally consistent under all processes $\mathbb{X}$ which admit a universally consistent online learner, hence they do not need to be consistent for *all* processes (this is impossible, by our Theorem 9); for such classes, our condition A describes precisely which processes $\mathbb{X}$ admit universally consistent learners (Theorem 11), and our condition B describes which concept classes $\mathcal{H}$ with infinite VCL trees admit optimistically universal learners (Theorem 12).
>
> We indeed discuss (and use, with citation) the results of the Bousquet et al paper referenced in the review.  The sufficiency half of our results for classes having no infinite Littlestone tree (Theorem 10) are directly based on that work.  We note that their work considers whether there exists an online learner successful under *arbitrary* realizable data sequences.  In contrast, we are interested in understanding which pairs ($\mathcal{H}$,$\mathbb{X}$) admit the existence of successful online learners, and the question of whether there can exist a learner which achieves this adaptively (i.e., without dependence on the distribution of the sequence): the former is the question of which processes admit universally consistent learners, and the latter is the question of which concept classes admit optimistically universal learners.  For instance, consider the class $\mathcal{H}$ of thresholds on the real line; this class admits an infinite Littlestone tree (hence Theorem 10 (which is similar to, but not quite the same as, Bousquet et al.'s Theorem 3.1) shows there is not an algorithm universally consistent under *all* realizable data sequences), but since there is no infinite VCL tree our Theorem 9 reveals that every process $\mathbb{X}$ admits a (process-specific) universally consistent online learner.  Hopefully this clarifies: the latter is about whether each process admits a (distribution-dependent) universally consistent learner, whereas the former is about whether there is an algorithm (defined independent of the process) which is universally consistent for every process $\mathbb{X}$.

---

> > ### Comment · Reviewer_jKRy · 2024-08-09
> >
> > Thank you for the clarification. I now better understand that: For classes that have an infinite VCL tree, the property of being an optimistically universal learner only requires the learner to be universally consistent under all processes which admit a universally consistent online learner. This resolves the apparent contradiction I had mentioned in my original review. As a consequence, I have increased my score. However, my reservation about the quality of the presentation remains.

---

### Official Review · Reviewer_vKfY · 2024-07-11

**Soundness:** 3
**Presentation:** 3
**Contribution:** 3
**Rating:** 7
**Confidence:** 3

**Summary:**

This paper considers characterizations for optimistically universal online learnability, a notion that refers to the existence of learning rules that are simultaneously optimal for any data-generating process that satisfies some minimal assumptions, i.e. admitting an optimal process-dependent learning rule. Previous works have studied this notion of learnability when the function being learned can be any measurable function, a very broad class of functions. In this paper, the authors consider characterizations of learnability when the labels of the samples are  realizable some known binary hypothesis class H, and also consider the agnostic task of achieving low regret with respect to the best hypothesis in H. They give characterizations of when H will be optimistically learnable based on VCL and Littlestone trees as well as some additional assumptions in certain cases. They provide clear examples to illustrate these characterizations.

**Strengths:**

I think the question of characterizing optimistic learnability for a given concept class is very interesting, and the paper is well-written in that it clearly explains the intuition and interpretations of the results, and connects to prior work.

**Weaknesses:**

I found some of the notation, especially around VCL trees and games, extremely difficult to parse. For instance, it was confusing that a large X may refer to either a tuple of X-values, or just a single X value. It would be great if this could be made more readable.

**Questions:**

- Do these results/techniques have any implication for optimistic online learning beyond the binary-label setting?

**Limitations:**

The authors have adequately address the limitations and assumptions made in their results.

---

> ### Author Rebuttal · Authors · 2024-08-05
>
> We appreciate your helpful comments.
>
> We will work to make the notation more readable, such as changing the capital X in the VCL game to make it less confusing.
>
> As for the optimistically universally online learnability problem beyond binary-labeled cases, this is a very interesting question.  We believe the technique used here extends easily to finite-label multiclass learning.  Beyond that, we have found that further research is needed to understand the problem for infinite-label multiclass learning, regression, and learning with general losses, and we leave this for future work.

---

> > ### Comment · Reviewer_vKfY · 2024-08-07
> >
> > Thanks for your reply and answers to my questions! I will keep my score as-is.

---

### Official Review · Reviewer_MwJp · 2024-07-12

**Soundness:** 4
**Presentation:** 2
**Contribution:** 3
**Rating:** 8
**Confidence:** 3

**Summary:**

This work continues the line of work by Hanneke and Blanchard et al universal online learning under general stochastic processes (instead of the more common iid or adversarial settings) with binary labels. While previous papers focused on the case of all measurable functions (as an implicit hypothesis spaces), this current work gives characterization for arbitrary hypothesis spaces.

The two main characterizations are as follows. Fix hypothesis space $H$.
1. For any data generating process $X$ there exists some algorithm that "learns" (i.e., average regret goes to $0$) on $X$ iff $H$ has no infinite VCL tree (a combinatorial dimension like VC used in universal learning).
2. For any data generating process $X$ there exists some algorithm that learns on $X$ and there exists a single algorithm that learns under any data process "under which learning is possible" iff it has no infinite Littlestone tree (a standard combinatorial dimension, characterizing online learning).

By the first chracterization, the existence of an infinite VCL tree just admits, thus, some data processes where learning fails, but it's unclear if learning still is possible on different specific processes. To tackle this, the authors offered an additional more fine-grained condition for each fixed (unlabeled) stochastic process $X$ (for the case the $H$ has infinite VCL trees):
1. $X$ is learnable iff a countable set of experts exist (which can depend on $X$) that satisfy a certain no-regret condition.
2. there is a single algorithm that learns "whenever learning is possible" (i.e., for any process that admits learning) iff there exists a countable set of experts (with no dependents on $X$) that satisfy the no regret condition for all processes $X$ that admit learning.

Both of these conditions collapse to a previously studied condition if $H$ is the set of all measurable functions.

**Strengths:**

Strong progress for an important and recent learning problem.

This work continues the characterization of distribution-dependent (universal) learning under stochastic processes, which fits into a broad range of recent results in universal learning and combinatorial characterizations.

The experts advice algorithm Squint is used as a nice black-box reduction from agnostic to realizable, similar to the weighted majority based reductions in adversarial online classification.

**Weaknesses:**

The paper lacks a bit of exposition to guide the reader and a more thorough discussion of related work.

**Questions:**

The paper only seems to discuss qualitative statements (learning possible iff no infinite tree,...). As far as I can see no quantitative rates are provided (like in uniform/universal learning). Are such rates, even very loose ones, possible (perhaps as future work)?

**Limitations:**

See questions.

---

> ### Author Rebuttal · Authors · 2024-08-05
>
> We appreciate your helpful comments. We will provide more discussion on related works in the final version and help the reader understand our results more easily.
> In our work, we are focus on the consistency problem. The quantitative rates problem is an interesting future direction, though will certainly require significant work even to formulate the question appropriately (for instance, it could be a question about, for a given rate function R(n), which function classes and processes admit learning with mistakes growing at rate R(n)?  It is definitely an interesting direction, though perhaps challenging, and worthy of a separate future work)

---

> > ### Comment · Reviewer_MwJp · 2024-08-12
> >
> > This is a strong and important paper. I think it should be accepted and hence I increase my score.

---

### Author Rebuttal · Authors · 2024-08-05

We want to thank all reviewers for their helpful suggestions and comments. We will polish our paper to make it easier to read and follow.

We here clarify our results again:

First, we would like to clarify the motivation of our work. This line of work on optimistically universal learning is trying to capture the minimal assumption needed for learnability. We investigate the **universal consistency** problem **beyond i.i.d. data-generating processes** and for general concept classes.

To emphasize our novelty and contribution: We provide a full characterization of what data processes admit universal online learning under concept class H and what concept classes H have an optimistically universal learning rule. The formal statements are in Theorems 9, 10, 11, 12, and for agnostic cases: Theorems 24, 25. Those results are all new results, except the sufficiency of Theorem 10, which can be derived from Bousquet et al. (2021).  The sufficiency of Theorem 9 is totally novel, though we use some techniques from Bousquet et al. (2021) and Alon et al. (2021).  The necessity of Theorem 10 is also (subtly) different from the necessity part of Theorem 3.1 in Bousquet et al (2021), since our data process must be fixed in advance (rather than adaptively constructed by an adversary) and we need to show not only that the number of mistakes is infinite but that it does not grow sublinearly.

---

### Decision · Program_Chairs · 2024-09-25

**Decision:**

Accept (poster)

**Comment:**

This work studies online learning under general stochastic processes (instead of the more common iid or adversarial settings) with binary labels continuing the line of work by Hanneke [2021]. While previous papers focused on the case of all measurable functions, this current work gives characterization for arbitrary hypothesis spaces. At a technical level the problem is a natural generalization of well-studied settings and the results are a significant progress. At the same time it is rather theoretical even by the standards of ML theory. In addition, the presentation is terse and hard to read even for experts on online learning. Still I think that the strengths of the results compensate for the limitations and recommend acceptance